# New Entity—Thalassemic Endocrine Disease: Major Beta-Thalassemia and Endocrine Involvement

**DOI:** 10.3390/diagnostics12081921

**Published:** 2022-08-09

**Authors:** Mara Carsote, Cristina Vasiliu, Alexandra Ioana Trandafir, Simona Elena Albu, Mihai-Cristian Dumitrascu, Adelina Popa, Claudia Mehedintu, Razvan-Cosmin Petca, Aida Petca, Florica Sandru

**Affiliations:** 1Department of Endocrinology, C. Davila University of Medicine and Pharmacy & C.I. Parhon National Institute of Endocrinology, 011684 Bucharest, Romania; 2Department of Obstetrics and Gynaecology, C. Davila University of Medicine and Pharmacy & University Emergency Hospital, 011684 Bucharest, Romania; 3Department of Endocrinology, C.I. Parhon National Institute of Endocrinology, 011684 Bucharest, Romania; 4Department of Dermatovenerology, C. Davila University of Medicine and Pharmacy & “Elias” University Emergency Hospital, 011684 Bucharest, Romania; 5Department of Obstetrics and Gynaecology, C. Davila University of Medicine and Pharmacy & “Filantropia” Clinical Hospital, 011684 Bucharest, Romania; 6Department of Urology, C. Davila University of Medicine and Pharmacy & “Prof. Dr. Theodor Burghele” Clinical Hospital, 011684 Bucharest, Romania

**Keywords:** beta-thalassemia, hypogonadism, fertility, pregnancy, bone, hypothyroidism, hypoparathyroidism, ferritin, cortisol, osteoporosis, diabetes

## Abstract

Beta-thalassemia (BTH), a recessively inherited haemoglobin (Hb) disorder, causes iron overload (IO), extra-medullary haematopoiesis and bone marrow expansion with major clinical impact. The main objective of this review is to address endocrine components (including aspects of reproductive health as fertility potential and pregnancy outcome) in major beta-thalassemia patients, a complex panel known as thalassemic endocrine disease (TED). We included English, full-text articles based on PubMed research (January 2017–June 2022). TED includes hypogonadism (hypoGn), anomalies of GH/IGF1 axes with growth retardation, hypothyroidism (hypoT), hypoparathyroidism (hypoPT), glucose profile anomalies, adrenal insufficiency, reduced bone mineral density (BMD), and deterioration of microarchitecture with increased fracture risk (FR). The prevalence of each ED varies with population, criteria of definition, etc. At least one out of every three to four children below the age of 12 y have one ED. ED correlates with ferritin and poor compliance to therapy, but not all studies agree. Up to 86% of the adult population is affected by an ED. Age is a positive linear predictor for ED. Low IGF1 is found in 95% of the population with GH deficiency (GHD), but also in 93.6% of persons without GHD. HypoT is mostly pituitary-related; it is not clinically manifested in the majority of cases, hence the importance of TSH/FT4 screening. HypoT is found at any age, with the prevalence varying between 8.3% and 30%. Non-compliance to chelation increases the risk of hypoT, yet not all studies confirmed the correlation with chelation history (reversible hypoT under chelation is reported). The pitfalls of TSH interpretation due to hypophyseal IO should be taken into consideration. HypoPT prevalence varies from 6.66% (below the age of 12) to a maximum of 40% (depending on the study). Serum ferritin might act as a stimulator of FGF23. Associated hypocalcaemia transitions from asymptomatic to severe manifestations. HypoPT is mostly found in association with growth retardation and hypoGn. TED-associated adrenal dysfunction is typically mild; an index of suspicion should be considered due to potential life-threatening complications. Periodic check-up by ACTH stimulation test is advised. Adrenal insufficiency/hypocortisolism status is the rarest ED (but some reported a prevalence of up to one third of patients). Significantly, many studies did not routinely perform a dynamic test. Atypical EM sites might be found in adrenals, mimicking an incidentaloma. Between 7.5–10% of children with major BTH have DM; screening starts by the age of 10, and ferritin correlated with glycaemia. Larger studies found DM in up to 34%of cases. Many studies do not take into consideration IGF, IGT, or do not routinely include OGTT. Glucose anomalies are time dependent. Emerging new markers represent promising alternatives, such as insulin secretion-sensitivity index-2. The pitfalls of glucose profile interpretation include the levels of HbA1c and the particular risk of gestational DM. Thalassemia bone disease (TBD) is related to hypoGn-related osteoporosis, renal function anomalies, DM, GHD, malnutrition, chronic hypoxia-induced calcium malabsorption, and transplant-associated protocols. Low BMD was identified in both paediatric and adult population; the prevalence of osteoporosis/TBD in major BTH patients varies; the highest rate is 40–72% depending on age, studied parameters, DXA evaluation and corrections, and screening thoracic–lumbar spine X-ray. Lower TBS and abnormal dynamics of bone turnover markers are reported. The largest cohorts on transfusion-dependent BTH identified the prevalence of hypoGn to be between 44.5% and 82%. Ferritin positively correlates with pubertal delay, and negatively with pituitary volume. Some authors appreciate hypoGn as the most frequent ED below the age of 15. Long-term untreated hypoGn induces a high cardiovascular risk and increased FR. Hormonal replacement therapy is necessary in addition to specific BTH therapy. Infertility underlines TED-related hormonal elements (primary and secondary hypoGn) and IO-induced gonadal toxicity. Males with BTH are at risk of infertility due to germ cell loss. IO induces an excessive amount of free radicals which impair the quality of sperm, iron being a local catalyser of ROS. Adequate chelation might improve fertility issues. Due to the advances in current therapies, the reproductive health of females with major BTH is improving; a low level of statistical significance reflects the pregnancy status in major BTH (limited data on spontaneous pregnancies and growing evidence of the induction of ovulation/assisted reproductive techniques). Pregnancy outcome also depends on TED approach, including factors such as DM control, adequate replacement of hypoT and hypoPT, and vitamin D supplementation for bone health. Asymptomatic TED elements such as subclinical hypothyroidism or IFG/IGT might become overt during pregnancy. Endocrine glands are particularly sensitive to iron deposits, hence TED includes a complicated puzzle of EDs which massively impacts on the overall picture, including the quality of life in major BTH. The BTH prognostic has registered progress in the last decades due to modern therapy, but the medical and social burden remains elevated. Genetic counselling represents a major step in approaching TH individuals, including as part of the pre-conception assessment. A multidisciplinary surveillance team is mandatory.

## 1. Introduction

Beta-thalassemia (BTH), a recessively inherited haemoglobin (Hb) disorder, is characterised by reduced or suppressed synthesis of β-globin chains with increased alpha/beta ratio. It is categorised into three main subtypes: major, intermedia and minor. Traditionally, BTH is highly prevalent in Mediterranean, Middle Eastern, and Southeast Asian countries; modern dynamics of the population increased the prevalence in North America, as well as Central and North Europe [1,2,3,4,5,6,7,8,9,10].

Worldwide, incidence is 0.3–0.4 million persons/year who are born with different types of haemoglobinopathies, thalassemia representing the most frequent hereditary anomaly of haemoglobin [1,2,3,4,5,6,7,8,9,10,11]. The damage of erythropoiesis induces premature erythrocytes destruction, causing chronic anaemia with extra-medullary haematopoiesis (EH) and associated bone marrow expansion (BME) in TH individuals (individuals diagnosed with major beta-thalassemia) [1,2,3,4,5,6,7,8,9,10]. Blood profile typically includes microcytic anaemia, and high or normal ferritin; Hb electrophoresis highlights the subtype of TH, and genetic testing confirms the disease, included as part of the prenatal protocol [1,2,3,4,5,6,7,8,9,10].

TH carriers are usually asymptomatic; major alpha TH is associated with a very high mortality at birth. In order to survive, patients with major BTH require lifelong transfusions beginning in early childhood; intermediate alpha and beta subtypes have heterogeneous presentations from severe to mild forms [1,2,3,4,5,6,7,8,9,10]. BTH-related morbidities are caused by EH, BME, and iron overload (IO) in the endocrine glands, skeleton, heart, liver, pancreas, and kidneys, etc. [1,2,3,4,5,6,7,8,9,10]. IO induces an excessive amount of reactive oxygen species (ROS) with consequent organ and vascular damage [1,2,3,4,5,6,7,8,9,10,11]. BTH persons might experience as a direct or indirect consequence of IO, EH, BME, and associated therapy: cardiac failure, pulmonary hypertension, pulmonary restrictive dysfunction, liver disease, renal dysfunction, leg ulcers, chronic pain, gallstones, thrombosis, psychiatric disorders, viral infections due to serial transfusions, and various skin manifestations, etc. [12,13,14,15]. Moreover, endocrine glands are particularly sensitive to iron deposits, hence the associated panel of hormonal imbalance is massive, representing a major challenge of the already complicated clinical course of the disease. The management strategies of TH include: transfusions, chelation agents against IO, hydroxyurea, splenectomy, stimulation of foetal Hb synthesis, and bone marrow/hematopoietic stem cell transplantation [1,2,3,4,5,6,7,8,9,10,16,17,18].

Generally, patients requiring transfusion and iron chelation therapy have a more severe clinical picture and overall outcome than those with mild/asymptomatic presentations [17]. Symptomatic individuals with intermedia TH are candidates for therapy, for instance, folic acid supplementation and splenectomy up to the complex panel of intervention that typically addresses major BTH [1,2,3,4,5,6,7,8,9,10]. The prognostic of TH has massively improved during last decades due to therapy progress, but the medical and social burden remains elevated, with numerous complications and hospitalizations. Genetic counselling represents a major step in approaching TH individuals, including as part of the pre-conception assessment. A multidisciplinary surveillance team is mandatory [1,2,3,4,5,6,7,8,9,10,18,19].

We introduce an update to current understanding of thalassemic endocrine disease (TED), presenting it as a complex picture of endocrine disorders (ED) associated with major BTH, including aspects of reproductive health including fertility potential and pregnancy outcome. Our main objective is to address endocrine components (altered or not) in major beta-thalassemia patients.

## 2. Methods

This is an overview of the literature; we included full-text articles, published in English, based on PubMed research; the timeline concerns papers published within the most recent 5 years (between January 2017 and June 2022). The keywords of research are “beta-thalassemia” in combination with one of the following: “endocrine”, “fertility”, respective “pregnancy”; but, also, “hypogonadism”, “thyroid”, “TSH”, “parathyroid”, “parathormone”, “stature”, “puberty”, “pituitary”, “diabetes”, “glycaemia”, “fracture”, “TBS”, “DXA”, “ACTH”, “osteoporosis”, “osteopenia”, or “adrenal”. Core endocrine descriptive analysis as displayed in Table 1 was restricted to clinical studies with different levels of statistical evidence, in both paediatric and adult population with major BTH, including more than 40 participants/study aiming two types of data: the ratio of EDs among the general panel of complications; and correlations between EDs and other specific parameters of evaluation in BTH (we included 1 longitudinal study, 15 cross-sectional studies, 1 retrospective analysis, 1 cohort study, 2 meta-analysis, and 2 surveys) [20,21,22,23,24,25,26,27,28,29,30,31,32,33,34,35,36,37,38,39,40,41] (Table 1).

## 3. Thalassemic Endocrine Disease

TED is a complex picture with multiple endocrine involvements in addition to the general, already complicated, picture of major BTH. The TED concept underlines well-known EDs which are particularly described in BTH patients since early age, and some of them demonstratean age-dependent pattern, such as thalassemia bone disease (TBD). A subchapter of TED includes reproductive health issues from the fertility profile to conceiving in order to deliver a healthy new born; recent progress in the management of major BTH in addition to progress inassisted reproductive techniques (ART) and genetic counselling helps TH women to conceive, hence it is important to address TED at highest standards (Figure 1).

Genotype–phenotype correlations showed that multi-organ IO (such as cardiac or pancreatic) is more frequent among homozygous individuals; however, TED is less predictable [42,43]. Not all studies clearly identified correlations between IO, serum ferritin values, imaging assessments of organ overload, iron chelation history, and splenectomy status with the degree of endocrine burden which should be expected in most cases, but the model of prediction concerning hormonal imbalance remains heterogeneous. General disease control is the most useful indicator of TED at any age. Endocrine glands are particularly sensitive to IO which is the key to understanding the TED spectrum [24,28,29]. TED is described at very young ages, for instance, one in four patients aged below 12 y already has at least one ED [24]. BTH-associated endocrine panel includes anomalies of GH/IGF1 (Growth Hormone/Insulin-like Growth Factor) axes with potential growth retardation, GH deficiency being identified in both children and adults, hypogonadism, hypothyroidism, various glucose profile anomalies, hypoparathyroidism, adrenal insufficiency, as well as reduced bone mineral quantity and the deterioration of microarchitecture [24,28,29]. Secondary complications of TED are expected. Diabetes mellitus (DM) is associated with well-known infectious and dermatologic complications, partly from anomalies of the cardio–metabolic profile [44]. Hypogonadism, either central or of gonadal causes, is manifested as pubertal delay, sexual dysfunction, fertility disturbances in adult females and males, as well as reduced bone mineral density (BMD) [24,28,29]. Children below the age of 12 y have been found with malnutrition (70%), but also at least 1 ED (23.33%) [24]. ED correlated with ferritin and poor compliance to therapy [24]. One study on N = 612 patients identified 40% with ED, partly from DM and osteoporosis (OP) [29]. Another study on N = 713 individuals found 86.6% with ED [33]. A meta-analysis indicated a prevalence of 43.92% for ED (non-DM) [34].

TED massively impacts the medical and social burden, including the quality of life [45]. Specific therapy for BTH might improve endocrine involvement by correcting underling mechanisms such as IO; however, not all the studies agree on specific BTH treatment-ED association. For instance, one study on 31 transfusion-dependent BTH patients (mean age of 16.9 ± 3.8 y, between 9 and 23 y) showed that endocrine involvement of 83% at deferasirox therapy initiation is reduced to 25.8% (*p* < 0.005) during an average follow-up of 5.9 ± 2.02 y (between 1 and 10 y) of chelation therapy [46]. A large multicentre, longitudinal study (median follow-up of 8 y, maximum of 18.5 y) on 426 subjects with transfusion-dependent TH in addition to long term therapy with deferasirox showed that, initially, 121/425 subjects had one ED, and 187/426 had at least 2 EDs. During follow-up, another 104 EDs were registered, thus the overall risk of a new ED is 9.7% within 5 y (95% CI 6.3–13.1). Age represents a positive linear predictor for ED (*p* = 0.005), as does TSH value (*p* < 0.001), Regardless of the type of ED, the number of EDs at baseline represents a negative linear predictor for another endocrine gland manifestation during surveillance (*p* < 0.001). Furthermore, deferasirox seemed to lower the risk of ED [28].

### 3.1. Growth Retardation and GH/IGF1Axis

Patients with major BTH experience low GH levels as part of TED [20]. Children with GH deficiency and short stature are candidates for GH replacement. The benefits of adult GH substitution therapy are debatable; currently, there are no specific randomised trials to address this issue in adults [47,48]. Low IGF1 in populations with GH deficiency is identified in 94.4% and 93.6%, respectively, on persons without GH deficiency. Short stature is related to low GH/IGF1 values, but not exclusively, as low IGF1 is also found in patients without GH deficiency [20].

A meta-analysis from 2021 included 74 studies concerning the growth status in patients with major BTH, between 1978 and 2019 (mean age of 14 y). The pooled prevalence of short stature was 48.9% (males more affected than females) and 26.6%, respectively, GH deficiency (95% CI 16-40.8) [25]. Half of the patients with a mean age of 14 y displayed different types of growth anomalies [25,27]. Another study suggested that short stature is the most frequent ED—a cohort on 24 persons with major BTH (mean age of 21.7 ± 8 y) showed that, independently of severe liver IO, 26% of them had short stature, followed by 16% with pre-DM, 14% with subclinical hypothyroidism, 14% with hypogonadotropic hypogonadism, and 12.5% with DM [49].

One study identified a higher percentage, noting that88% of children with major BTH suffered from short stature [22]. This is an observational study from Central India which included 50 children (mean age of 15.98 ± 3.4 y, ranges between 8 and 18 y) between 2014 and 2016. In addition to short stature, delayed puberty was found in 71.7%, hypothyroidismin 16%, and DM in 10%. Serum ferritin correlated with TSH (Thyroid Stimulating Hormone), glycaemia, and pubertal status delay [22].

The assessment of GH stimulation tests might explain differences among studies; dynamic assays are an essential part of TED management. One cross-sectional, multicentre study from 2022 evaluated GH/IGF1 axis on 81 adults with major BTH (44/81 males, average age of 41 ± 8 y) who were undergoing treatment with transfusion and chelation. The patients took a GHRH (GH Releasing Hormone) + arginine test; 18 subjects were diagnosed with a GH deficiency during the test, not at baseline. This subgroup had a higher body mass index (BMI) and a more severe lipid profile (*p* < 0.05), but with similar liver function when compared to BTH individuals without GH deficiency [20].

IGF1 production may be independent of GH control, for instance, in the relationship between liver status and IO [20]. Other than correlations with serum ferritin, we also identified transferrin saturation (TS) and EDs in the study [23]. A single centre, transversal study on 58 adults with transfusion-dependent BTH were associated with growth retardation, evaluated IO, ferritin levels, and TS, as well as thyroid function and IGF1. The subjects (53.4% males, median age of 21, between 18 and 24 y) demonstrated a 32.7% prevalence of subclinical hypothyroidism, of which a respective 79.3%Pdisplayed low IGF1. TS correlated with FT4 (free levothyroxine), respective with IGF1 (r = −0.361, *p* = 0.003, respectively, r = −0.313, *p* = 0.008), not with TSH, neither at multivariate regression considering FT4, TSH and IGF1 [23].

Since the patients with intermediate BTH have different clinical presentations, EDs vary, and a subgroup of subjects was similar to patients with major BTH with regards to the profile of EDs [1,2,3,4,5,6,7,8,9,10,11,12,13,14,15,16,17,18,19,20,21,22,23,24,25,26,27,28,29,30,31,32,33,34,35,36,37,38,39,40,41,42,43,44,45,46,47,48,49,50]. Overall, short stature affects both children and adults; GH/IGF1 should be promptly explored in the young population due to the benefits of early GH replacement.

### 3.2. Hypogonadism in BTH

TED-related hypogonadism is due to IO, general health status, BTH complications, and potential side effects of different regimes. It manifests during puberty or adult life, either as primary or secondary amenorrhea in females [51,52]. Its prevalence varies among studies, related especially to whether or not the pubertal status is taken into consideration. Long-term hypogonadism induces a high cardiovascular risk in addition to toxic effects of cardiac IO [53]. According to one study, the prevalence is 23.6% among patients at puberty age, and ferritin negatively correlates with pituitary volume [27]. Some suggested that puberty failure might be predicted by serum ferritin [22,52]. A transversal study from a single tertiary centre included 58 patients (33/58 males) with transfusion-dependent TH (aged between 17 and 19 y) and 72.4% of subjects experienced either normal puberty or delayed onset with spontaneous progression, but 26.7% of them experienced arrested puberty requiring hormonal intervention. Multivariate regression identified serum ferritin as the single parameter correlating with pubertal failure/arrest (OR of 1.005, 95% CI of 1.002–1.009) [26].

We mention several studies in which hypogonadism was reported with the highest percentage among other EDs [31,33,38,41]. We will now provide some prevalence data on different cohorts. For instance, a retrospective, single centre study on 45 adults with transfusion-dependent TH (22/45 males; mean age of 28.8 ± 6.9 y; 71.1% with major BTH) revealed that 54% had at least one ED, 38.9% had two EDs, and 11.1% had three EDs or more [31]. The most frequent ED was hypogonadism (22.2%), followed by OP (20%), hypothyroidism (13.3%), DM (6.7%), and hypocortisolism (4.4%). Ferritin was not correlated with EDs [31]. A large study from 2017 on 613 transfusion-dependent BTH patients (54.3% males, mean age of 13.3 ± 7.7 y) identified the following ratio of complications: 76.4% for heart events, 46.8% for hypogonadism, 22% for hypoparathyroidism, 8.3% for hypothyroidism, 7.3% for DM, and a respective 1.8% for kidney dysfunction. Hypogonadism was the most frequent complication in participants below the age of 15, while cardiac events were for people older than 15 y [41]. Another cohort identified the percentage of hypogonadism to be 44.5% (N = 713) [33]. A study on 280 patients with transfusion-dependent, major BTH identified the prevalence of hypogonadism as 82%, stunting as 69%, hypoparathyroidism as 40%, and hypothyroidism as 30% [38]. The sensitivity of hypogonadism in order to predict severe myocardial siderosis is 90%, probably a surrogate marker in case T2* MRI (Magnetic Resonance Imaging) is not available [38].

### 3.3. BTH-Related Hypothyroidism

TED-associated hypothyroidism rates (either clinically manifested or subclinical, either primary or secondary) depend on study, criteria, population, disease control and applied therapy for BTH [21,54,55]. Ultrasound screening of the thyroid is not mandatory, since anomalies of thyroid function are easily found [56]. Thyroid disease is found not only in individuals with transfusion-dependent BTH, but in patients who suffered a transplant, too [55]. Adequate chelation treatment might improve thyroid anomalies.

One study on 82 persons with major BTH treated with deferoxamine identified 29.27% of cases with subclinical hypothyroidism, 1.22% with overt form; non-compliance to iron chelator treatment increased the risk of thyroid anomalies 6.38-foldversuscompliant subjects (relative risk of 6.386; 95% CI 2.4–16.95), suggesting that adequate chelation treatment might reduce the burden of thyroid conditions [32]. Another study on 120 children younger than 12 y who were diagnosed with major BTH undergoing long-term protocols of transfusion and chelation treatment showed the most common ED to bethyroid disease (9.17% of patients), followed by glucose profile anomalies (7.5%), which meant 4.17% of all children had impaired glucose tolerance (IGT), a respective 3.33% had impaired fasting glucose (IFG), and none had franc DM. The rarest endocrine comorbidity was hypoparathyroidism (6.66%). The risk factors for ED included: high ferritin (OR of 0.98; 95% CI of 0.96–0.99, *p* = 0.003), poor compliance with BTH therapy (OR of 0.38; 95% CI of 0.16–093, *p* = 0.03), and the use of combined chelating agents lowers the risk of endocrinopathies versus single agent use (*p* = 0.04) [24].

However, not all studies agree concerning chelation influence on thyroid dysfunction. A Malaysian study published in 2021 included 51 transfusion-dependent TH individuals (47% males; 68.8% had major BTH). A total of 21.6% of subjects had hypothyroidism, and 63.6% of them had a central mechanism. Furthermore, 27.3% had synchronous hypogonadism, 9.1% had hypogonadism and DM, and 9.1% had central hypocortisolaemia. Hypothyroidism was not correlated with age, ferritin, history of iron chelation or splenectomy status [21].

Most authors consider subclinical forms to be more frequent [21,54]. One study from 2018 on 83 children with major BTH (older than 3 y; 59% males) identified a ratio of 4.8% concerning subclinical hypothyroidism; TSH was not correlated with serum ferritin, neither with the profile of oral chelation and transfusions [37]. The predominant mechanism is central (yet not unanimously recognised) [21,54]. A transversal study on 67 patients with major BTH (mean age of 15.37 ± 3.73 y) identified a 10.4% rate of hypothyroidism; all the cases were subclinical as defined by TSH > 6.5 mIU/L and T4 > 4.2 ng/dL, and no case of overt or central hypothyroidism was identified. Ferritin positively correlated with TSH (*p* = 0.008), but not with T4; ferritin was higher in persons with BTH and thyroid dysfunction versus those with normal thyroid function [35].

We conclude that hypothyroidism is mostly pituitary-related; it is not clinically manifested in the majority of cases, hence the importance of TSH/FT4 screening. It is found at any age, with a prevalence that varies between 8.3% and 30% [38,41]. Paediatric ratio is between 4.8 and 16% [22,37]. According to some studies, hypothyroidism is not correlated with age, ferritin, or splenectomy status; others showed ferritin-TSH or TS-FT4 correlations, while some authors identified a positive influence of chelation therapy [21].

The pitfalls of TSH interpretation due to hypophyseal IO should be taken into consideration at the initial diagnostic of hypothyroidism and during the follow-up under T4 replacement. Some authors observed the reversibility of the condition under adequate IO reducing chelation regime [57].

### 3.4. Major THD-Associated Hypoparathyroidism

TED-related hypoparathyroidism is caused by BTH-related IO at the level of parathyroid glands. The rate of the condition varies. Children with major BTH below the age of 12 y had 6.66% hypoparathyroidism in one study [24]. The prevalence of hypoparathyroidism is identified as: 13.2% (N = 713), 40% (N = 280), 22% (N = 613) [33,38,41].

In BTH patients with hypoparathyroidism, serum ferritin might act as a stimulator of Fibroblast Growth Factor 23 which controls the serum levels of 1,25-dihydroxyvitamin D3 values [58]. Anomalies of klotho were suggested as well, but there is still a gap in terms of clinical evidence [59]. An ICET-A (International Network of Clinicians for Endocrinopathies in Thalassemia and Adolescence Medicine) survey from 2018 included 3023 persons with major BTH and 739 with intermediate BTH; the results showed that 6.8% and 4.4% of them, respectively, had hypoparathyroidism (onset age between 10.5 and 57 y and between 20 and 54 y, respectively); associated hypocalcaemia varied from asymptomatic to severe manifestations, and hypoparathyroidism is mostly associated with growth retardation and hypogonadism among EDs in major BTH (53% and 67% of cases, respectively) [40].

The patients without transfusion necessary might have similar parathyroid-related complications with transfusion-dependent TH, but less often [40]. VD supplementation in different formulas is essential; the choice of VD therapy also depends on VD deficiency (VDV) as reflected by serum 25-hydroxyvitamin D which is reported as part of TBD (see Section 3.7).

### 3.5. Adrenal Gland Status in Patients with Major BTH

TED-adrenal gland dysfunction is mild and a certain index of suspicion should be kept into consideration during surveillance due to potential life threatening complications. Some authors recommend periodic check-up of ACTH (Adrenocorticotropic Hormone) stimulation test thus a level of awareness in completely asymptomatic patients is mandatory because they might underline a chronic partial adrenal insufficiency, either primary or secondary, as part of TED.

One cross-sectional study on 72 adults with major BTH identified that 20% of them have some degree of adrenal dysfunction based on serum and salivary cortisol levels during 1 µg ACTH stimulation test [39].

Adrenal insufficiency/hypocortisolism status is the rarest ED in most studies (but some studies reported a prevalence of up to one third of patients) [57]. This is a cohort of 28 participants with major BTH (16/28 males) in which growth retardation was identified in 57% of them (13/16 with delayed puberty). GH deficiency was found in 35% of cases, while adrenal insufficiency was confirmed in 32% of cases; 21% of participants also had subclinical hypothyroidism, and 35% of patients had low BMD. Overall, 82% of patients had at least one ED [57].

Another bias comes from the fact that many analyses did not routinely perform a dynamic test in order to detect adrenal function anomalies, hence a suboptimal diagnostic might be considered. An ICET-A 2019 survey concerning occult endocrine complications of major BTH enrolled 3.114 adults from 15 countries, of which a respective 202 subjects were younger than 18 y; the rates of central hypothyroidism, GH deficiency and latent cortisol deficiency were of 4.6%, 3%, and 1.2%, respectively, and 0.5%, 4.5%, and 4.4% in the paediatric population [36].

Atypical EM site might be found in adrenals [60]. We mention an interesting case of incidentaloma: a 24-y-old female was detected with an adrenal mass that was confirmed as being an adrenal EM due to a previously unrecognised form of BTH [61]. Another case of adrenal EM was published in 2022 (a 40-y-old female with local pain due to a right adrenal tumour of 5.8 cm) [62]. The authors reviewed the literature and identified a total of 14 cases of adrenal EM [62]. This particular aspect in patients with thalassemia mimics an incidentaloma scenario [60,61,62].

### 3.6. Glucose Profile in Major BTH

Understanding the chronic IO-associated effects of hypoxia, inflammation and increased oxidative stress represents a step forward concerning the pathogenic panel of EDs. The glucose profile deterioration in major BTH is time-dependent due to pancreatic and peripheral (especially hepatic) IO, thus glycaemia-related TED is one of the most dynamic elements over time [63].

Serum ferritin, IO, and markers of oxidative stress might predict glucose anomalies, meaning impairment of insulin pancreatic secretion, but also peripheral tissue insulin resistance [64,65]. Duration of BTH negatively impact glucose homeostasis [66]. Damage of glycaemia control in addition to pancreatic IO are risk factors for cardiovascular complications, and anomalies of lipids profile [67,68]. An original study published in 2020 evaluated the adipokines profile in 30 participants with transfusion-dependent (major) BTH (group 1) versus 30 persons with minor BTH (group 2) versus 20 controls (healthy subjects). Group 1 had statistically significant lower levels of leptin (versus group 2, respective of controls), higher adiponectin (versus controls), and higher resist in in any group with thalassemia when compared to healthy group. Leptin negatively correlated with ferritin in group 1 [69].

The complex frame of adipokines anomalies might underline some of the BTH-associated EDs, especially glucose anomalies and bone status. A prospective, small sample study of seven individuals with transfusion-dependent BTH (mean age 22.4 ± 4.2 y) showed that after initial normal oral glucose tolerance test (OGTT) but with hypoinsulinemia, the patients developed glucose profile anomalies within 43 ± 26 months (between 11 and 80 months), as follows: 2/7 individuals were diagnosed with IGT, 3/7 had concomitant IGT and IFG, and 2/7 had DM (this subgroup had at baseline the lowest insulinogenic index) [70]. This study proves progressive insulin response deterioration as the hallmark of TED-associated glucose anomalies.

Insulin secretion-sensitivity index-2 might represent a surrogate marker of beta-pancreatic cell dysfunction as reflected during OGTT, a useful indicator to be followed in individuals with major BTH [71]. A continuous glucose monitoring system might highlight early glucose profile anomalies in children with major BTH [72]. It is indicated for a selective high-risk subgroup of patients [72]. Glucose anomalies in major BTH include IGF, IGT and DM [73,74,75]. Annual screening of glucose profile potential damage starts by the age of 10 y [73,74,75]. OGTT is a useful method, but it is more difficult to assess when compared with simple fasting glycaemia [73,74,75]. HbA1c is less practical in this disorder, but it may be used; fructosamine and glycated albumin represent some potential alternatives [73,74,75].

Pancreatic MRI in addition to insulin secretion-sensitivity index-2 represents a promising alternative to be implemented in daily practice in the future [73]. One meta-analysis from 2019 focused on glycaemia profiles in major BTH; 44 studies were included (N = 16,605 patients); the rate of DM was 6.54% (95% CI 5.3–7.78), IFG of 17.21% (95% CI 8.43–26), IGT of 12.46% (95% CI 5.98–18.94) versus non-DM EDs of 43.92% (95% CI 37.94–49.89) [34]. The previously mentioned larger studies found a higher percentage for DM: 7.3% (N = 613), 15.9% (N = 713), 34% (N = 612 patients) [29,33,41]. A paediatric study showed that 10% of individuals with major BTH have DM, and ferritin correlated with glycaemia [22]. Another study on a cohort of children aged below 12 y identified that 7.5% of them were associated with glucose profile anomalies [24]. Many studies do not take into consideration subgroups with IFG, IGT or do not routinely perform OGTT, which might explain different results.

### 3.7. Thalassemia Bone Disease

Skeletal health is affected by hypogonadism-related OP, renal function anomalies, glucose profile imbalance, GH deficiency, malnutrition, chronic hypoxia-induced calcium malabsorption, and transplant-associated medication, with consecutive increased fracture risk, all of these pathogenic elements being contributors to TBD which has an early onset, developing a continuously increasing risk of fragility fractures during lifespan [76,77].

Genetic influence concerning collagen or vitamin D (VD) receptor polymorphism has a potential contribution to TBD [78]. The degree of disease control and associated treatments might influence TBD as well, but there is a multifactorial panel. For example, one study from 2021 in adults with TBD highlighted a correlation between BMD and TS, not serum ferritin [79]. Another study in children older than 6 y identified suboptimal BMD in 74% and 86%, respectively, (femoral neck and lumbar BMD, respectively) of 50 patients with major BTH, and in this cohort, BMD correlated with pre-transfusion Hb [80]. Murine experiments confirmed the contribution of chronic kidney disease, including high phosphate, to TBD [81].

Overall prevalence of fractures in thalassemia patients is higher than that of the general population. We mention a meta-analysis from 2021 including 25 studies (N = 4934 subjects with different types of TH) that point out the pooled prevalence of fractures is 17% for major BTH, and 18% for transfusion-dependent TH, respectively, higher than non-transfusion-dependent TH (7%), and alpha TH (4%) [82].

Exclusion of liver analysis due to high IO provides an adequate result in individuals with poorly controlled BTH when using central lumbar DXA (Dual-Energy X-ray Absorptiometry) [83]. Despite being the “gold” standard for BMD evaluation, particular microarchitecture and anatomic effects of IO represents pitfalls of DXA, especially in children, persons with growth retardation and severe spine deformities due to vertebral fractures (VF) [77,84].

Low BMD was identified in both the paediatric and adult population; the prevalence of OP/TBD in major BTH patients varies, the highest is 40–72% (depending on age, studied parameters, DXA assessments and corrections, and screening thoracic–lumbar spine X-ray for VF). [77,85,86]. We mention a study on 177 adults with major BTH compared to 490 normal subjects who were age-matched (between 20 and 39 y, participants of Iranian Multi-centre Osteoporosis Study) [87]. The authors performed a calibration method to the reference population which is essential in order to provide adequate results for young individuals with BTD. Thus, the initial ratio of low spine Z-score ≤−2 of 56% was re-calculated to 72% [87]. A study on 29 adults with major BTH, aged between 23 and 44 y, identified low BMD in 42.9% of females, 23.1% of males, respectively; the rates of prevalent fractures were 26.7% and 35.7%, respectively. The patients with hypothyroidism (which generally is not a risk factor for osteoporotic fractures) had a higher prevalence of low BMD (*p* = 0.016) [85]. A large cohort was a retrospective UK study on 612 participants with transfusion-dependent BTH (between 2009 and 2018) showed that 76% of the individuals had at least one complication (54% has at least two, 37% had at least three), and among them, 40% had OP, and 34% had DM [29].

As mentioned, patients with intermediate BTH have various rates of EDs, some subgroups being similar with major BTH. A cross-sectional, multicentre study on 522 subjects with intermediate BTH (average age of 30.8 ± 12.1 y) identified EDs as following: 22% of participants had osteoporosis/osteopenia, 10% had hypogonadism, and 5% had primary hypothyroidism. Multivariate regression showed age as risk factor for low BMD (splenectomy was also a risk factor for OP), hypogonadism, and DM without insulin requirement (the use of hydroxyurea seemed protective against this form of DM). These data represent the largest study on intermediate BTH focusing on EDs [88]. Another cohort study from 2019 included 713 persons with transfusion-dependent BTH (aged between 10 and 62 y); the results showed that 86.8% of them had at least one ED, the most frequent being low BMD (72.6%), followed by hypogonadism (44.5%), DM (15.9%), hypoparathyroidism (13.2%), and hypothyroidism (10.7%). Age, splenectomy status and BMI correlated with ED [33].

TBS (Trabecular Bone Score), a practical tool to assess bone quality, has been used for TBD, and we mention two studies [89,90,91]. TBS is particularly useful in diabetic patients [91]. A cross-sectional study on TBD included 86 patients aged ≥18 y with different haemoglobinopathies; 16.3% was the prevalence of radiographic evidence concerning VF and all of them had *β*-thalassemia/Hb*E*. Low BMD, reduced TBS and the presence of ED were significantly associated with VFs [89].

Moreover, advanced methods of BTH control increased the number of the aging population with associated new endocrine concerns such as age-dependent risk of osteoporotic fractures [92,93]. BTD prevalence is higher in older patients with major BTH, especially regarding VFs with negative impact on quality of life due to local pain and vertebral deformity. A cross-sectional, single centre study among 82 patients with major BTH revealed that 35% had vertebral deformities with lower TBS versus people without VFs (1.141 ± 0.083 versus of 1.254 ± 0.072, *p* < 0.0001). TBS was a better discriminator of VF that BMD and DXA Z-score [90]. An increased prevalence of non-traumatic fractures versus the general population has been reported among young persons with major BTH. One study on 179 patients (105/179 males, aged between 3.6 and 28.3 y) identified that 21% of them had long bones fractures, and a prevalence of 4.5% for VF, an equivalent of 307 fractures per 10,000 patient-years [94].

In addition to DXA, peripheral quantitative computerised tomography has been used in children with TBD. We mention a case-control study on 334 children (167 with major BTH and 167 controls, aged between 3 and 18 y) that confirmed a higher prevalence of VF and long bone fractures (*p* < 0.05) in BTH, a lower areal BMD, reduced muscle area and decreased cortical thickness (*p* < 0.05), but higher trabecular densities and lumbar apparent BMD which may be caused by IO and erythroid hyperplasia [95].

Bone turnover markers might be helpful in daily clinical practice. Their assessment in children after hematopoietic stem cell transplantation should reflect the changes of the general clinical status; bone formation markers such as osteocalcin are expected to increase [96]. A study of 62 subjects with major BTH (32 males/30 females) showed that RANKL (receptor activator of nuclear factor kappa-Β ligand) was higher (*p* = 0.049) and OPG (osteoprotegerin)/RANKL ratio was lower (*p* = 0.009) in patients with BTH versus a group of menopausal osteoporotic women, with similar results for sclerostin and CTX (C-terminal telopeptide), which suggests that traditional medication against OP should not be different in TBD [97]. A study on adults with major BTH showed a 10-fold increase in serum sclerostin versus controls, being correlated with pre-transfusion Hb levels, LIC (Liver Iron Concertation), splenectomy status, and prevalent fractures [98]. Another study from 2022 confirmed lower OPG, a respective lower OPG/RANKL ratio in 60 children with transfusion-dependent BTH (aged between 5 and 14 y) versus 60 age—and sex- matched healthy subjects, with a higher incidence of OPG rs2073618 (×2.3) and OPG rs2073617 genotypes (×1.9) among TBD [99]. Dysfunction of ORG/RANKL axis has been identified in patients with BTH intermedia, as well [100,101,102]. Bone turnover markers might reflect the transitory hemodynamic anomalies concerning transfusion-associated Hb changes.

VDD had been found in relationship with kidney status anomalies, calcium intestinal absorption impairment, and malnutrition [76]. A systematic review from 2021 identified 12 articles on VDD among the thalassemia group, concluding that VDD prevalence and its severity varies among different studies, while it is debatable whether routine VD supplementation might improve BMD under these circumstances [76]. One case-controlled study found that patients with major BTH have an 8-fold increase in VDD, probably with an important contribution of hepatic hemosiderosis [102,103].

As the therapy of choice for secondary OP, we mention bisphosphonates and denosumab; as primary prevention, the hormonal replacement therapy for hypogonadism is helpful [104,105]. One randomised, placebo-controlled, phase 2b study on denosumab (N1 = 32 patients treated with 60 mg denosumab every 6 months versus N2 = 31 individuals with placebo, for 1 year) evaluated for the first time in subjects with transfusion-dependent TH noggin levels, an antagonist of bone morphogenetic protein as promotor of bone formation. Noggin is statistically significantly higher in both N1 and N2, but in N2 it increases more than in N1, suggesting a different intervention of denosumb [106].

From what we know so far, the best therapy option in order to increase BMD and prevent osteoporotic fractures in major BTH is similar to general population. A meta-analysis from 2018 on nine randomised, controlled studies on bisphosphonates showed that mostly zolendronate, but also alendronate, seem to have an impact on TBD [107]. Particular aspects are related to the long-term effects of anti-osteoporotic medication on bone dynamics in young patients and females of reproductive age.

There is also the issue of whether the iron chelation regime is adequate for preventing BTD. A non-randomised study on 256 patients with transfusion-dependent BTH followed BMD-DXA for 1 year under five different chelation therapies with similar results on BMD changes as well as serum ferritin [108]. We identified a single study from 2022 on teriparatide (real life experience), as bone anabolic agent for severe OP, in 11 adults with TBD (male/female ratio of 6/5, mean age of 45 ± 4.38 y). The therapy was followed for 19 ± 7 months; the main causes of withdrawal were side effects (5/11) and poor compliance to therapy; the results showed significant increase in BMD, respective of serum osteocalcin, and cross-laps [109]. The drug seems promising in TBD; however, the level of statistical evidence is low and its administration is restricted to specific countries protocols.

### 3.8. Imaging of Endocrine Glands in BTH

Various imaging techniques are useful in TED in addition to hormonal assays. In BTH, imaging assessments embrace particular aspects due to IO. As mentioned, musculoskeletal evaluation is important in children and adults, yet there are some other particular aspects [110]. Pituitary hemosiderosis in children with BTH might predict delayed puberty, short stature and central hypothyroidism [22,111,112,113]. Interestingly, brain MRI data showed conflicting results regarding IO; noting that the pituitary gland is particularly sensitive to iron deposits following the trend of the other endocrine glands [112]. An MRI-based study on 40 patients with transfusion-dependent TH identified that iron deposits are positive in 62.5% of persons in the liver, 45% in the pancreas, and 12.5% in the heart. Three out of fifty patients had at least one ED [113]. The cardiac T2* sequence evaluating iron upload negatively correlates with fasting glycaemia (*p* = 0.03); subjects with short stature had increased cardiac iron overload (*p* = 0.01). Those with hypothyroidism had higher hepatic overload (*p* = 0.05), and females displayed increased pancreatic iron burden T2* sequence when compared to males [113].

Another 3T MRI study was performed in 57 subjects with major BTH versus 30 controls; 56.1% of the patients had short stature (defined as height below the third percentile for age); 23.4% of the pubertal subgroup had hypogonadism (the cut-off for delayed puberty definition was lack of pubertal development ≥13 y for females, ≥14 y for males); ferritin negatively correlates with pituitary volume (*p* = 0.006); anterior pituitary volume is lower in subjects with hypogonadism (*p* = 0.012). Overall, the study suggests that 3T MRI-based pituitary volume might be predictive for IO-related hypogonadism [27].

An MRI study on 50 participants with transfusion-dependent TH identified at least one ED in 2/3 patients; hypogonadism and DM are not correlated with a higher pituitary IO; while hypogonadism correlates with cardiac MRI-based IO (*p* = 0.004); short stature correlates with hepatic IO (*p* = 0.05) [30]. LIC quantification performed with MRI scans seems an alternative tool to asses control disease status [114]. LIC was evaluated on 52 patients with major BTH and found to be correlated with IGF1 levels, as well as serum ferritin values [115]. The introduction of oral iron chelation therapy by the end of puberty improves the prevalence of ED, as well as LIC results [116].

One case report was published in 2021 presenting an autopsy study of a 10-y-old boy with major BTH associating multi-endocrine involvement by IO; a hypophyseal null-cell microadenoma was identified which was not prior described in patients with THB [117]. At this moment, we consider this diagnostic to be incidental; we do not have imaging data to confirm a higher risk of pituitary incidentalomas, but, indeed, the patients more often have imaging scans performed due to surveillance protocols.

## 4. Fertility Issues in Females and Males Diagnosed with Major BTH

As mentioned, the rates of hypogonadism varies among subjects with major BTH, as major part of TED [118]. This ED may be an isolated endocrine disturbance, or it accompanies anomalies of the thyroid, parathyroid glands, glucose profile and skeleton status. Other elements of TED like severe hypothyroidism/myxedema and complicated DM might supplement the impact on the reproductive potential. On the other hand, persistent, untreated hypogonadism impairs the achievement of peak bone mass in teenagers and young adults, and contributes to adult OP [119,120].

IO-related anterior pituitary damage causes hypogonadotrophic hypogonadism. Significantly, imaging studies might predict the pituitary damage-related hypogonadism due to particular anatomic configuration of iron uptake, also a potential reversibility concerning pituitary lesions has been suggested through iron chelators’ use. MRI studies might help with the distinction between normal pituitary function and central hypogonadism [121,122].

An ovarian lesion might co-exist with hypogonadotropic hypogonadism due to similar underling mechanisms [123]. The theory of iron gonadal toxicity is sustained by low Anti-Müllerian Hormone (AMH) and other ovarian reserve markers [123]. Uysal A et al. showed that secretion of AMH, FSH, LH, and antral follicle count (AFC), as well as estradiol and ovarian volume, is significantly decreased in women with major BTH [123]. Direct effects of iron on ovarian function are part of the complicated puzzle concerning females’ fertility in major BTH. Similarly, Talaulikar SV. et al. published a longitudinal study on the effect of IO over ovarian reserve in 17 women with transfusion-dependent (major) BTH who were followed for 10 y versus 52 age-matched healthy controls. AMH, AFC and estradiol were found significantly lower in BTH females (*p* < 0.05) [122].

Hormonal replacement therapy is necessary in cases with BTH-related hypogonadism in addition to specific therapy for underling disease, and it seems safe in both females and males [124]. Early iron chelation therapy improves IO, thus the damage of endocrine glands like pituitary and gonads which are responsible of the hypogonadism [125]. Gonadotropin replacement might help arrested puberty in major BTH [126]. Studies strictly concerning testosterone substitution in males with BTH are less substantial. We identified one study published in 2019 on 95 men with major BTH under testosterone for hypogonadism; 43.1% of them experienced gynecomastia, local skin reaction at injection or patch was reported in one of three subjects [127].

Females with BTH have a high risk of infertility and premature ovarian failure, independently or synchronous with pituitary-related hypogonadism [128,129]. Ovarian tissue and oocyte cryopreservation are part of modern fertility strategies, and it should be taken into consideration in young females with major BTH, as a promising alternative in the field of reproductive health [128]. Deep-frozen procedures allow back transplantation of ovarian tissue several years since early removal during childhood at young adults who are further referred to in vitro fertilization [129].

Males with BTH are at risk of infertility due to germ cell loss [118]. IO also induces an excessive amount of free radicals which impairs the quality of sperm, iron being a local catalyser of ROS [130,131]. The first MRI study on testes (2017) showed that, in patients with transfusion-dependent BTH, MRI T2 values are lower versus healthy control, and correlate with serum ferritin; IO-related testicular damage explains infertility in males without hypogonadotropic hypogonadism [132]. Another study showed that male fertility is lower in patients (aged between 16 and 41 y) who had undergone hematopoietic stem cell transplant (N = 43) compared to those treated with transfusion and chelation (N = 52). The prevalence of hypogonadism (32%) was similar among the two groups, confirming that male fertility is a consequence of non-hormonal anomalies [133].

In order to preserve spermatogonial stem cells, recent experimental data highlighted the potential use of testicular tissue (TT) banking programs, a procedure which is recommended for young patients with different malignancies before starting oncologic therapies. TT cryopreservation is still under development; it involves a surgical technique with an irreversible outcome, as far as we know for the moment; the practical application in males with major BTH is to be further determined. Probably, a selected sub-group of high-risk individuals will benefit from this new method of fertility preservation [134]. Analysing the spermatogonial cells in the testes after testicular biopsy might represent a future option in male fertility by developing new techniques of achieving post-meiotic cells in vitro [135].

## 5. Pregnancy Outcome

Since major BTH is associated with a high rate of infertility, there are limited data concerning spontaneous pregnancies, most of patients requiring an inductor of ovulation, and, eventually, a method of ART in order to achieve a pregnancy [136]. Due to the advance of current therapies, the reproductive health of females with major BTH is improving, and “hundreds of pregnancies have been reported so far” [137] (Figure 2).

Treatment of major BTH is expensive and difficult for any affected person, and preventive programs should be available to BTH individuals independently whether they live in areas with limited economical resources or not [138,139,140]. The pre-conception evaluation is mandatory, including partner screening for BTH as well as spermogram, viral tests, and risk factors for thrombophilia [141,142]. Next-generation sequencing (NGS) of pre-implantation genetic testing for alpha and beta-thalassemia is a safe and accurate test; it can be synchronously performed with aneuploidy screening [143,144].

A promising alternative is represented by non-invasive prenatal diagnostic based on isolated cell-free DNA from maternal blood, a test whose accuracy is yet to be determined [145]. Another future option is suggested by the assessment of several blood protein biomarkers (such as serotransferrin) from mothers with BTH that might indicate whetheror not the foetus is affected by major BTH [146]. Genetic therapies are under development, but they are not currently a practical approach [147].

Amniocentesis is performed between weeks 16 and 18 of pregnancy; if the foetus is affected, abortion cannot be performed in reasonable time. If amniocentesis is offered before week 14, there is a high risk of complications considering the new-born. An alternative method to amniocentesis is transabdominal chorionic villus sampling (TA-CVS) which seems safe for women with major BTH [148]. This procedure can be performed earlier, at weeks 10–12 of gestation [149,150].

For instance, we mention a very large Italian trial from Sardinia where the prevalence of thalassemia is high: during a period of 40 years, a total of 8564 foetal procedures concerning prenatal diagnostic were performed via TA-CVS (2138 foetuses with BTH). Of these, 92.8% of couples chose termination of the pregnancy. In order to avoid this, pre-implantation genetic diagnostic was proposed and performed in 184 cases with a rate of pregnancy success between 11% and 30.8% (initially, this procedure was allowed only for infertile couples) [148].

A retrospective study on 467 foetuses in at-risk pregnancies coming from the Hakka population included prenatal genetic diagnosis, which combined genomic DNAs analysis from villus, amniotic fluid or cord blood to peripheral blood of the parents (a total of 88 CVS samples, and 375 amniocentesis procedures) including 111 patients with BTH (27.93% of them with major BTH); the method provided a reliable result concerning prenatal diagnostic [151]. A secondary technique, such as NGS or Sanger sequencing, might improve the diagnostic [144,151].

In addition to hypogonadism, multi-organ damage impairs fertility potential and pregnancy outcome, but also, maternofoetal risks may be caused by medication for different co-morbidities in major BTH, inclusive of post-transplant immunosuppressive medication [152,153]. Early hematopoietic stem cell transplant during childhood might prevent puberty failure and even conserve fertility [154]. A complete assessment of IO, heart function by electrocardiography and echocardiography, and ultrasonography of the liver are mandatory pre-conception. For instance, one case of a female with major BTH who had spontaneous pregnancy was admitted for multiple complications due to transfusion-dependent disease, including myocardial siderosis, and OP which was treated with oral alendronate. She was unintentionally under therapy with hydroxyurea, deferiprone and bisphosphonate until week 20 of the pregnancy, which was complicated by growth restriction and oligohydramnios and terminated by premature delivery (*via* caesarean), but no major effects were identified in the healthy new-born [155].

Potential pregnancy complications in BTH mothers are: preeclampsia, gestational DM and high blood pressure, pulmonar hypertension, thrombosis, kidney failure, poly- and oligo-hydramnios, placenta praevia, and pleural effusion [156].

Moreover, the effect of chronic anaemia and diffuse IO might dramatically impact pregnancy outcome [157,158]. Anaemia represents a complication of minor BTH, but also iron-deficient anaemia is expected during physiological pregnancies [158].

One study included 53 women with transfusion-dependent BTH (French National Registry) compared to 37 healthy controls, between 1995 and 2015. The results showed the following: 5/53 were twin pregnancies, ART was applied in 9/53 pregnancies, the median delivery term was 39 weeks (similar with control group), and 53.6% had a caesarean procedure.The panel of significant pregnancy-related complications include: six cases of each thrombosis, severe infections, gestational hypertension, intrauterine growth restriction; five cases with severe haemorrhages; and four with gestational DM, etc. Remarkably, four cases of postpartum OP were identified within the first year after birth. Infant weight at birth was also statistically significantly reduced versus controls [159].

One observational study on 60 pregnant women with haemoglobinopathies (81% with BTH) showed that thyroid pathology was the most frequent ED; intrauterine growth restriction and placenta praevia were also common; overall, there were 57 live births and one stillbirth, but no major obstetric complications were found in BTH [160].

Transfusion support during pregnancy aims a Hb level above 10 g/dL to help achieve adequate foetus growth. Treatment with iron chelators should be discontinued from the first trimester of pregnancy. Resumption of chelation is indicated after birth. Desferrioxamine is potentially beneficial at the end of the second trimester, and is applicable in selected cases [137].

Fozza C et al. published a study on pregnancy outcome in females with major BTH. The mean age of the 15 women was 33 y (ranges between 28 and 38 y). A total of 7/33 patients underwent spontaneous pregnancies and 9/33 patients underwent induced ovulation with gonadotrophin, and intrauterine insemination was used in 3/33 cases. Iron chelators were stopped during pregnancies. In total, 79% of patients experienced a successful delivery. Among the complications of pregnancy were gestational diabetes (N = 2), and one case of intrauterine foetal death [161].

Glycaemic status should be monitored in pregnant women with BTH. HbA1c is an indicator of Hb glycosylation within the most recent 3 months. Haemoglobinopathy affects the value of HBA1c leading to misinterpretation of the glucose profile [162]. Zhang X. et al. investigated whether the HBA1c values in combination with gestational DM patients were affected by minor BTH. This case-control study included 41 patients with gestational DM and minor BTH, 93 patients with gestational DM without BTH (control group). Blood glucose was not statistically significantly different between the groups (*p* < 0.05); HbA1c was significant lower in the BTH group (*p* < 0.05). HbA1c is not a good indicator for monitoring blood glucose in pregnant women with BTH [163].

Patients with major BTH were associated with a hypercoagulability status in addition to endothelial damage secondary to formation of iron-mediated free radicals and in association with the 3–4-fold increase in thrombosis risk during a pregnancy [137].

Despite the level of statistical evidence we have so far, pregnancy is not contraindicated in females with major BTH. An individual decision should be taken extremely cautiously by a multidisciplinary team. The complete panel of potential complications that interfere with pregnancy outcome also includes endocrine components from DM for the adequate replacement of hypothyroidism and sufficient VD to maintain bone health [164,165,166,167]. Prior anti-OP medication has insufficient evidence, but this is not a contra-indication [165,166,167]. Mild EDs such as subclinical hypothyroidism or IFG/IGT might become overt or clinically symptomatic during pregnancy, which is why practitioners should be aware of these issues.

## 6. Conclusions

Endocrine glands are particularly sensitive to iron deposits, and this particular aspect is the key TED picture which massively impacts the overall presentation and prognostic, including the quality of life for those with major BTH, hence the importance of endocrine awareness. The outcome in major BTH has registered massive progress in the last decades due to modern therapy, but the medical and social burden remains elevated. Genetic counselling represents a major step in approaching TH individuals, including as part of the pre-conception assessment. A multidisciplinary surveillance team is mandatory.

TED is a complex panel of EDs which might be registered since early childhood or with an age-dependent pattern. The endocrine elements may improve, or even remit, in association with control of underling disease. Most endocrine glands might be targeted in BTH with various levels of severity. Awareness and management of the endocrine panel is a direct contributor to the increased quality of life and general health status of these patients.

## Figures and Tables

**Figure 1 diagnostics-12-01921-f001:**
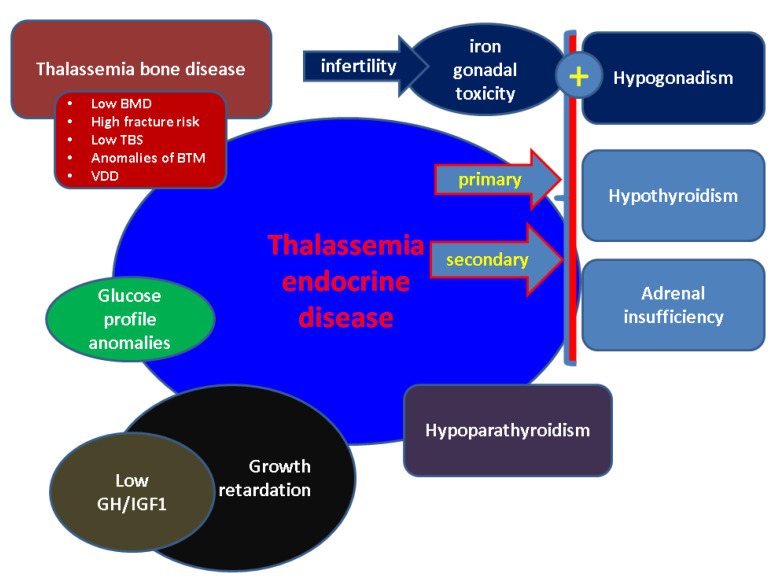
Overview of thalassemia endocrine disease (see references in text). Abbreviations: BMD = Bone Mineral Density, TBS = Trabecular Bone Score; BTM = Bone Turnover Markers, VDD = vitamin D deficiency; GH = Growth Hormone; IGF1 = Insulin-like Growth Factor.

**Figure 2 diagnostics-12-01921-f002:**
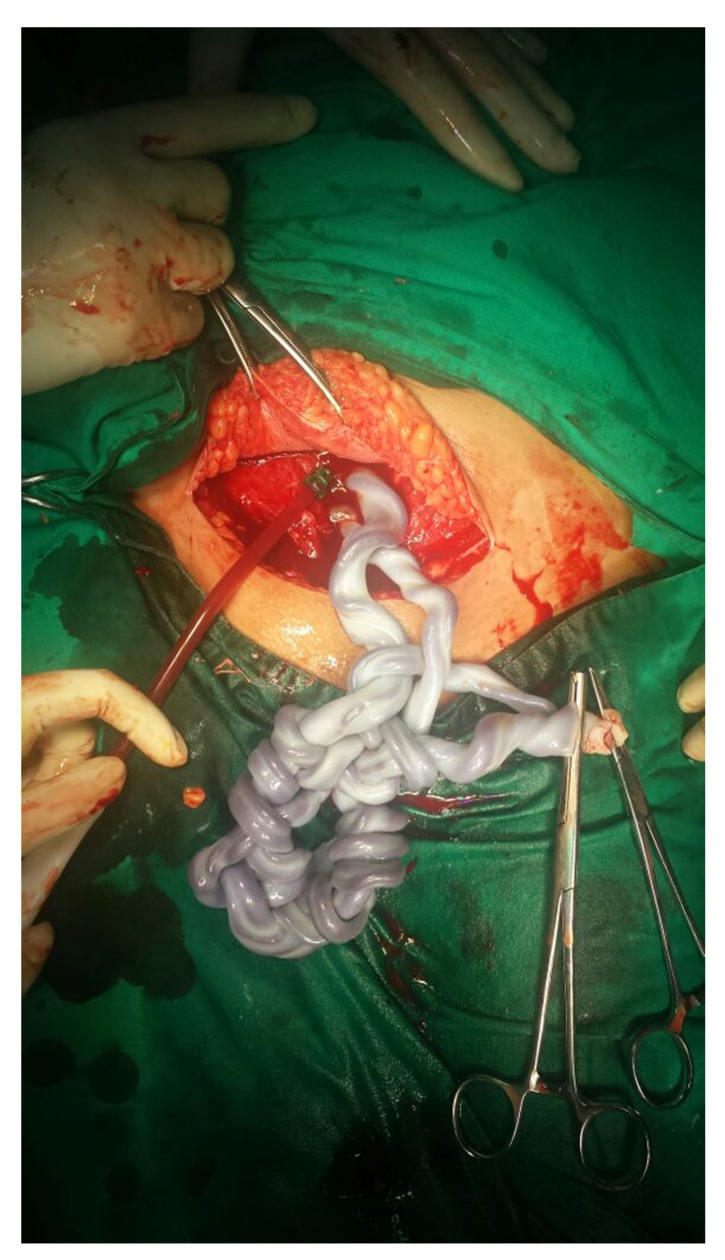
This is a previously unpublished case of a 37-year-old female with major beta-thalassemia (both of her parents were confirmed with minor beta-thalassemia); she was under a protocol of 2 Units RBC transfusions every two weeks; she developed post-transfusion chronic hepatitis C and had a splenectomy performed 5 years prior. She was admitted for secondary amenorrhea which was confirmed to be a spontaneous, monochorionic-monoamniotic twin pregnancy. During pregnancy she had increased liver enzymes three times above the normal limit and maintained a level of haemoglobin above 10 g/dL under protocol of transfusion. Both foetuses had normal development until week 31 when she suffered spontaneous ruptures of membranes. Caesarean section was performed with extraction of the foetuses (pelvic presentation) who were both heathy new-borns. Here is post-extraction capture: the two umbilical cords were wrap around each other.

**Table 1 diagnostics-12-01921-t001:** Core endocrine descriptive analysis of thalassemic endocrine disease. (We selected studies with more than 40 patients, published between 2022 and 2017, with data concerning the ratio/prevalence of endocrine disorders among patients with major beta-thalassemia and potential correlations of endocrine involvement with specific parameters of underling disease or general health. The display is based on publication’ year starting with the most recent [20,21,22,23,24,25,26,27,28,29,30,31,32,33,34,35,36,37,38,39,40,41].

First Author Publication Year Reference Number	Study Design	Results: Endocrine Involvement	Results: Endocrine Parameters Correlations Other Observations
Casale M. 2022 [28]	multi-centre, longitudinal follow-up median of 8 y	N = 426 patients with TST At baseline: 121/425 with 1 ED 187/426 with at least 2 EDs During follow-up: another 104 EDs Overall risk of a new ED = 9.7% within 5 y (95% CI 6.3–13.1).	Age is a positive linear predictor (*p* = 0.005); so is TSH (*p* < 0.001) for an ED The number of EDs at baseline is a negative linear predictor for another ED during follow-up (*p* < 0.001) Deferasirox ↓ risk of ED
agliardi I. 2022 [20]	cross-sectional, multi-centre	N = 81 adults with BTH major (44/88 males, mean age of 41 ± 8 y) Evaluation: GHRH + arginine test N1 = 18/81 with GH deficiency N2 = 63 without GH deficiency BMI, cholesterol N1 > N2 (*p* < 0.05) Liver function similar in N1 versus N2 Low IGF1 in N1: 94.4%; N2: 93.6%	Low IGF1 has multiple mechanisms, not only GH
Seow CE. 2021 [21]	cross-sectional, single centre	N = 51 patients with TDT (47% males; 68.6% with major BTH) 21.6% with hypothyroidism (63.6% with central hypothyroidism)	Most often type of hypothyroidism is central Hypothyroidism is not correlated with age, ferritin, splenectomy status, chelation therapy
Dixit N. 2021 [22]	observational (transversal)	N = 50 children with major BTH (mean age of 15.98 ± 3.4 y, between 8–18 y) 88% with short stature 71.7% with delayed puberty 16% with hypothyroidism 10% with DM	Ferritin correlates with TSH, glycaemia, and pubertal delay
Atmakusuma TD. 2021 [23]	cross-sectional, single centre	N = 58 adults with transfusion-dependent BTH + growth retardation (53.4% males, median age of 21, between 18 and 24 y) 32.7% with subclinical hypothyroidism 79.3% with low IGF1	TS correlates with FT4, respective IGF1, not with TSH
Mahmoud RA. 2021 [24]	cross-sectional	N = 120 children with major BTH (age < 12 y) 70% with malnutrition 23.33% with ED 9.17% with thyroid disease 7.5% with glucose profile anomalies 6.66% with hypoparathyroidism	Most common ED is at thyroid Endocrine involvement risk correlates with high ferritin, and poor compliance to BTH therapy
Arab-Zozani M. 2021 [25]	meta-analysis	N = 74 studies (mean age of 14 y) 48.9% = pooled prevalence of short stature (males more affected than females) 26.6% with GH deficiency	Half of patients have different growth anomalies
Singh P. 2021 [26]	cross-sectional, single centre	N = 58 patients with TDT (33/58 males, age between 17 and 19 y) 72.4% with normal puberty/delayed onset with spontaneous progression 26.7% with arrested/failure puberty	Multivariate regression identifies serum ferritin to correlate with pubertal failure/arrest
Nayak AM. 2021 [27]	cross-sectional (3T MRI)	N1 = 57 patients with major BTH versus 30 controls 56.1% with short stature 23.4% of the pubertal subgroup with hypogonadism	Ferritin negatively correlates with pituitary volume Anterior pituitary volume is lower in subjects with hypogonadism.
Jobanputra M. 2020 [29]	retrospective cohort	N = 612 patients with TDT 40% with 1 ED (non-DM) 40% with osteoporosis 34% with DM	10-y mortality rate of 6.2%
Karadag SIK. 2020 [30]	cross-sectional (MRI)	N = 50 patients with TDT 2/3 with at least 1 ED	Hypogonadism and DM do not correlate with pituitary IO. Hypogonadism correlates with cardiac MRI-based IO (*p* = 0.004) Short stature correlates with hepatic IO (*p* = 0.05)
Lee KT. 2020 [31]	retrospective, single centre	N = 45 adults with TDT (22/45 males; mean age of 28.8 ± 6.9 y; 71.1% with major BTH) 54% with at least 1 ED 38.9% with 2 EDs 11.1% with 3 or more EDs EDs: hypogonadism (most frequent, 22.2%), osteoporosis (20%) hypothyroidism (13.3%) DM (6.7%) hypocortisolism (4.4%)	Ferritin is not correlated with ED
Yassouf MY. 2019 [32]	cross-sectional	N = 82 patients with major BTH treated with deferoxamine 29.27% with subclinical hypothyroidism, 1.22% with overt hypothyroidism	Non-compliance to deferoxamine increases the risk of thyroid anomalies by 6.38-foldversus compliant subjects (RR of 6.386; 95% CI 2.4–16.95)
Bordbar M. 2019 [33]	cohort	N = 713 patients with TDT (aged between 10 and 62 y) 86.8% with at least 1 ED 72.6% with low BMD 44.5% with hypogonadism 15.9% with DM 13.2% with hypoparathyroidism 10.7% with hypothyroidism	Age, splenectomy status and BMI correlates with ED
He LN. 2019 [34]	meta-analysis	N = 44 studies N = 16,605 patients with BTH 6.54% with DM (95% CI 5.3–7.78) 17.21% with IFG (95% CI 8.43–26) 12.46% with IGT (95% CI 5.98–18.94 43.92% with non-DM EDs (95% CI 37.94–49.89)	Highest prevalence of DM (7.9%, 95% CI: 5.75–10) correlates with Middle East region
Baghersalimi A. 2019 [35]	cross-sectional	N = 67 patients with BTH (mean age of 15.37 ± 3.73 y) 10.4% with subclinical hypothyroidism	Ferritin positively correlated with TSH (*p* = 0.008), not with T4 Ferritin is higher in persons with thyroid dysfunction versus normal thyroid function
De Sanctis V. 2019 [36]	ICET-A survey	N1 = 3.114 adults with BTH N2 = 202 younger than 18 y with BTH 4.6% and 0.5%, respectively, with hypothyroidism 3% and 4.5%, respectively, with GH deficiency 1.2% and 4.4%, respectively, with latent hypocortisolism	The prevalence of occult EDs varies within different age groups
Upadya SH. 2018 [37]	cross-sectional	N = 83 children with major BTH (59% males, age ≥ 3 y) 4.8% subclinical hypothyroidism	TSH is not correlated with serum ferritin, oral chelation and transfusions
Ehsan L. 2018 [38]	cross-sectional	N = 280 with TDT 82% with hypogonadism 69% with stunting 40% with hypoparathyroidism 30% with hypothyroidism	The sensitivity of hypogonadism to predict severe myocardial siderosis is 90%
Ambrogio AG. 2018 [39]	cross-sectional	N = 72 adults with major BTH 20% adrenal dysfunction based on short Synacthen stimulation test	
De Sanctis V. 2018 [40]	ICET-A survey	N1 = 3023 patients with major BTH N2 = 739 patients with intermedia BTH 6.8% and 4.4%, respectively, with hypoparathyroidism onset age between 10.5 and 57 y, respectively, between 20 and 54 y	Hypoparathyroidism is associated mostly with growth retardation and hypogonadism in major BTH (53% and 67%, respectively, of cases)
Yaghobi M. 2017 [41]	cross-sectional	N = 613 patients with TDT (54.3% males, mean age of 13.3 ± 7.7 y) 46.8% with hypogonadism 22% with hypoparathyroidism 8.3% with hypothyroidism 7.3% with DM	Hypogonadism is the most frequent complication below the age of 15, and cardiac events are, for people older than 15 y

Abbreviations: BTH = beta-thalassemia; BMI = Body Mass Index; BMD = Bone Mineral Density; DM = diabetes mellitus; GH = Growth Hormone; GHRH = GH Releasing Hormone; ED = endocrine disease; FT4 = free levothyroxine; IGF1 = Insulin-like Growth Factor; IGT = impaired glucose tolerance; IGF = impaired fasting glucose; ICET-A = International Network of Clinicians for Endocrinopathies in Thalassemia and Adolescence Medicine; IO = iron overload; MRI = magnetic resonance imaging; RR = relative risk; TDT = transfusion-dependent thalassemia; TSH = Thyroid Stimulating Hormone; TS = transferrin saturation; y = year.

## Data Availability

Not applicable.

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
