# Peer review of "New Entity—Thalassemic Endocrine Disease: Major Beta-Thalassemia and Endocrine Involvement"

_diagnostics, 2022, doi:10.3390/diagnostics12081921_

Round 1

Reviewer 1 Report

The title of the manuscript can be changed. It is not clear what authors are proposing (a new nomenclature for an already defined disease or a new entity?).

The manuscript should be reorganized, better defining the actual objective of the review. I believe that the main objective is to address endocrine components (altered or not) in major beta thalassemia patients. If this is the case, both title and Abstract can be rewritten.

In fact, the Abstract (which is too long and confusing) seems to propose to group several diseases as a new entity. Nevertheless, as mentioned before, those factors are quite diverse even in non clinical patients (the authors themselves acknowledge that “prevalence of each ED varies with population, criteria of definition, etc.”)

Please define TH (thalassemic patients?)

At section 3 (Thalassemic endocrine disease) it is not clear the mention to assisted reproductive techniques (ART). Are authors planning to discuss genetic counseling directed to BTH patients? This does not seem to be an objective of this review.

 At section 5 (Pregnancy outcome), some data about pregnancy in BTH patients is given, but Figure 1, and the “unpublished case” report are misplaced there. Why are such Figure and case report doing there?

Extensive review of the manuscript is needed concerning English-language usage. As a general rule, data is presented as a “list” with few or no connections between different phrases and paragraphs. Frequently, sentences mix up different ideas (see, for example At section 5: “The treatment of major BTH is expensive and difficult for people in areas with limited economical resources [138]. Therefore, preventive programs like screening, prenatal diagnosis, a selective termination of pregnancy should be extremely carefully taken into consideration through a multidisciplinary decision for each individual/couple.” – I would say that “treatment of major BTH is expensive and difficult for” any affected person, and “preventive programs” should be available to BTH individuals independently if they live in “areas with limited economical resources” or not. What exactly is the point authors are intending to bring up?

In conclusion, the manuscript should clearly state its objectives, and should be revised in order to present data in an ordered way, not just as a list of citations.

Author Response

Dear Reviewer,

The title of the manuscript can be changed. It is not clear what authors are proposing (a new nomenclature for an already defined disease or a new entity?).

We changed the title. Indeed, the underling endocrine diseases in major beta thalassemia may be regarded as a distinct chapter in endocrinology which is important for different practitioners, not only endocrinologists. Thank you very much.

The manuscript should be reorganized, better defining the actual objective of the review. I believe that the main objective is to address endocrine components (altered or not) in major beta thalassemia patients. If this is the case, both title and Abstract can be rewritten.

We confirm the main objective. Thank you, we introduced the main objective as you suggested, both in the article (aim section), and in the abstract. Also, we reorganized the abstract and reduced its length. Thank you.

In fact, the Abstract (which is too long and confusing) seems to propose to group several diseases as a new entity. Nevertheless, as mentioned before, those factors are quite diverse even in non clinical patients (the authors themselves acknowledge that “prevalence of each ED varies with population, criteria of definition, etc.”)

We reduced and reorganized the Abstract. The entire panel of potential endocrine comorbidities that are related to major beta thalassemia we called “thalassemic endocrine disease” in order to provide practical insights of this numerous chapters. The mentioned endocrine anomalies may be presented since early age and they may remit or relapse depending of specific therapy for thalassemia and general health of a patient. Thank you for pointing out this aspect.

Please define TH (thalassemic patients?)

We confirm. We specified in the paragraph of Introduction. Thanks

At section 3 (Thalassemic endocrine disease) it is not clear the mention to assisted reproductive techniques (ART). Are authors planning to discuss genetic counseling directed to BTH patients? This does not seem to be an objective of this review.

Indeed, ART is not the objective of the manuscript. We only mention it in order to sustain the importance of recognizing and treating the associated endocrine diseases since the major step forward in ART helps the patients with beta-thalassemia to successfully achieve a pregnancy. We specified it in the first paragraph of section 3 and rephrased it. Thank you

At section 5 (Pregnancy outcome), some data about pregnancy in BTH patients is given, but Figure 1, and the “unpublished case” report are misplaced there. Why are such Figure and case report doing there?

This is not a case report. It is not unusual that in addition to general data of a review to provide an example like an imagery or a histological capture to highlight the general information, from authors’ daily experience. Here we introduced for practical points an original capture in addition to the general data we presented concerning the reproductive health in females who are planning to conceive, a female with beta major thalassemia at the moment of cesarean section after delivering two healthy twins.

Extensive review of the manuscript is needed concerning English-language usage. As a general rule, data is presented as a “list” with few or no connections between different phrases and paragraphs. Frequently, sentences mix up different ideas (see, for example At section 5: “The treatment of major BTH is expensive and difficult for people in areas with limited economical resources [138]. Therefore, preventive programs like screening, prenatal diagnosis, a selective termination of pregnancy should be extremely carefully taken into consideration through a multidisciplinary decision for each individual/couple.” – I would say that “treatment of major BTH is expensive and difficult for” any affected person, and “preventive programs” should be available to BTH individuals independently if they live in “areas with limited economical resources” or not. What exactly is the point authors are intending to bring up?

Thank you. We reorganized and revisited the manuscript’s sections. We reorganized the paragraphs. You marked with yellow color the changes or the parts where we either corrected or removed text. We reviewed the language. Also, we introduced the sentence concerning the fact that BTH comes with a major medical, social and economic burden, as you suggested.

In conclusion, the manuscript should clearly state its objectives, and should be revised in order to present data in an ordered way, not just as a list of citations.

Thank you very much. We reorganized the manuscript following the objectives.

Thank you very much

Reviewer 2 Report

The manuscript is well written and the topic is Novel 

The review is extensive discussing multiple endocrine disorders in thalassemia but it is better to shorten the text as much as possible and to replace the text with more tables and figures that may summarize the ideas of the text making it more easy to read 

Author Response

Dear Reviewer,

The manuscript is well written and the topic is Novel. The review is extensive discussing multiple endocrine disorders in thalassemia but it is better to shorten the text as much as possible and to replace the text with more tables and figures that may summarize the ideas of the text making it more easy to read.

Thank you. We re-organized the manuscript and introduced a Figure to highlight the endocrine panel which is associated with major beta thalassemia.

Thank you very much

Reviewer 3 Report

The article is very interesting. It is well designed and structured, containing the most relevant bibliography on the subject in the last 5 years. However, I think the article could be improved:

1 - Insert an image that describes the pathology

2 - I do not understand the relevance of figure 1

3 - The conclusion is very poor, and could be explored further.

Author Response

Dear Reviewer,

The article is very interesting. It is well designed and structured, containing the most relevant bibliography on the subject in the last 5 years. However, I think the article could be improved:

Insert an image that describes the pathology

Thank you. We did.

I do not understand the relevance of figure 1

For practical purposes we introduced a demonstrative capture of a female patient with major beta thalassemia who delivered two healthy new born babies. A review paper may be associated with images that highlight the importance of general data, and this was our choice concerning this particular matter due to the rarity of the situation.

The conclusion is very poor, and could be explored further.

Thank you. We added a second paragraph. “TED is a complex panel of EDs that might be registered since early childhood or with an age –dependent pattern. The endocrine elements may improve in association with control of underling disease, or even remit. Most of endocrine glands might be targeted in BTH with various levels of severity. Awareness of endocrine panel and adequate management is a direct contributor to increased quality of life and general health status of these patients.”

Thank you very much

Round 2

Reviewer 1 Report

Authors adequately approached all questions previously raised by this reviewer. Two points, but related to manuscript form, not scientific content should be referred: a) I still think that the Abstract section is too long, but I believe that this should not hinder the publication. b) Also,  I am not quite sure about the necessity to include the Figure 2, although I do understand the authors  explanation ("...an example like an imagery or a histological capture to highlight the general information, from authors’ daily experience...").

Anyway, I believe that Editors of the Journal should approach these points.